# Self-supervised contrastive learning unveils cortical folding pattern linked to prematurity

Julien Laval                                                   JULIEN.LAVAL@CEA.FR
Denis Rivière                                                  DENIS.RIVIERE@CEA.FR
Aymeric Gaudin                                                 AYMERIC.GAUDIN@CEA.FR
Vincent Frouin                                                 VINCENT.FROUIN@CEA.FR
Jessica Dubois                                                 JESSICA.DUBOIS@CEA.FR
Andrea Gondova                                        ANDREA.GONDOVA@ETU.U-PARIS.FR
Jean-François Mangin                                  JEAN-FRANCOIS.MANGIN@CEA.FR
Joël Chavas                                                    JOEL.CHAVAS@CEA.FR
*NeuroSpin, CEA Saclay, Université Paris-Saclay, France*

**Editors:** Under Review for MIDL 2024

## Abstract

Brain folding patterns have been reported to carry clinically relevant information. The brain folds mainly during the last trimester of pregnancy, and the process might be durably disturbed by preterm birth. Yet little is known about preterm-specific patterns. In this work, we train a self-supervised model (SimCLR) on the UKBioBank cohort (21070 adults) to represent the right superior temporal sulcus (STS) region and apply it to sulci images of 374 babies from the dHCP database, containing preterms and full-terms, and acquired at 40 weeks post-menstrual age. We find a lower variability in the preterm embeddings, supported by the identification of a knob pattern, missing in the extremely preterm population. The code can be found on Github.

**Keywords:** Brain, sulci, preterms, self-supervised learning, contrastive learning

## 1. Introduction

Brain folding can be described using sulci, which are the grooves separating surface humps, called gyri. Main sulci are common to all individuals, but they differ in shape. Recurrent shape patterns can be identified, and some have been reported to be correlated to pathologies (Mellerio et al., 2015; Kabat and Król, 2012). Therefore, studying folding patterns can pave the way for better diagnosis and understanding of brain diseases. In particular, finding preterm-specific patterns could lead to a better understanding of their developmental trajectories and their susceptibility to neurodevelopmental diseases (Hee Chung et al., 2020). In this paper, we apply a self-supervised contrastive learning algorithm based on SimCLR (Chen et al., 2020) to extract preterm-specific patterns.

## 2. Methods

**The dHCP database**: We process a subset of 374 subjects of the Developing Human Connectome Project (dHCP) database (Edwards et al., 2022) containing subjects born between 23 and 41 post-menstrual weeks and classified according to their degree of prematurity (Table 1). All images were acquired at $\sim 40$ weeks post-menstrual using the same machine and protocol.

| Degree of prematurity | number of subjects |
|---|---|
| Extremely preterm ($< 28$ weeks) | 19 |
| Very preterm ($28 \leq$ weeks $< 32$) | 35 |
| Moderate to late preterm ($32 \leq$ weeks $< 37$) | 43 |
| full-term ($\geq 37$ weeks) | 277 |

Table 1: Distribution of birth ages in dHCP, according to WHO classification

**Inputs:** The starting points are structural MR images of the brain. They are processed through the BrainVisa Morphologist pipeline[1] that skeletonizes a negative cast of the brain. It transforms the sulci into surfaces (3D objects of one-voxel width) following the middle of the folding. In practice, we use regional crops so that the algorithm can focus on specific sulci using the deep_folding toolbox (Chavas et al., 2022) [2]. Cortical skeleton images are affinely normalized in the Talairach space with a 2m̃m voxel size. This paper focuses on the right superior temporal sulcus (STS). It was chosen because its development spans the whole third trimester of pregnancy (Dubois et al., 2008).

**Model parameters and evaluation:** Gaudin et al. (2024) optimized a SimCLR pipeline applied to the cingular region of the brain, classifying the presence of the paracingular sulcus as a pretext task. They selected the backbone, the augmentations, and hyperparameters. The main augmentation is called "branch clipping". It removes branches of the skeleton until a given percentage of voxels is removed. No further optimization was performed in the present study because we want to evaluate the performance (thus the generalizability) of a model optimized on another brain region[3]. To assess the model performances, we apply a two-layer linear Support Vector Classifier to the model embeddings of dHCP for each prematurity group (the negative class being the full-terms) and measure the 3-fold cross-validation AUC (with class-based stratification).

## 3. Experiments and results

We train the model on the STS crops of 21070 adult brains from UkBioBank (Sudlow et al., 2015). Next, we compute the embeddings of the dHCP subjects and perform classification (§2) to assess performances. We reach an AUC of **87.4(1.0)** for the extremely preterm class (Fig. 1a), confirming the ability of the model to capture information related to prematurity. Using UMAP (McInnes et al., 2020), we visualize the embedding space. The more premature the babies, the less variable they seem to be (see Laval (2024) for the UMAP). We quantify it in the 10-dimension latent space and find significant loss of variability for extremely and very preterms compared to full-terms (Fig. 1b). Then, we focus on the extremely preterms to characterize the variability drop visually. We select the top-predicted full-terms because they are more likely to contrast with preterms, and we identify an STS pattern shared by all top-predicted subjects, which is in stark contrast with typical extremely preterm STS: a long sulcus with a deep large knob at its anterior extremity. This pattern is then

---

1. https://brainvisa.info

2. github: neurospin/deep_folding

3. The main parameters are the following: backbone: ConvNet, input size: $(1, 22, 49, 46)$, embedding size: 10, number of trainable parameters: 452 172, loss temperature: 0.1, drop rate: 0.05, branch clipping percentage: 40%, batch size: 16, lr: 0.0004, weight decay: 5.0e-05, max epochs: 250.

identified in at least 5/19 randomly selected full-terms, which confirms quantitatively that it is missing in the extremely preterm population ($\mathbf{p = 0.008}$ using Barnard's exact test, Fig. 1c).

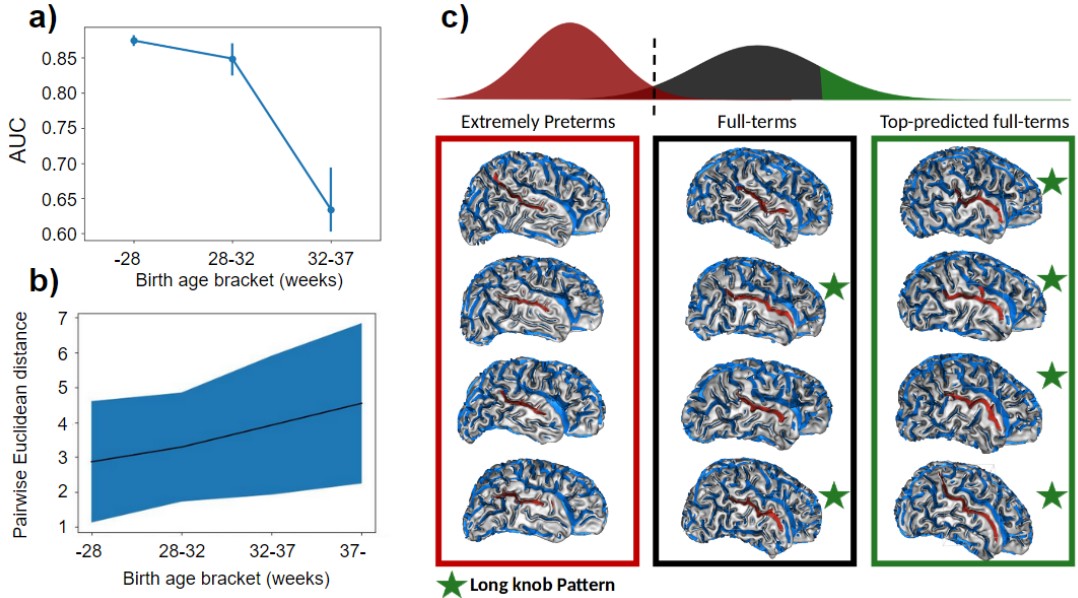

Figure 1: **a)** AUC between preterms and full-terms for each class of prematurity (averaged over 3 trainings). **b)** Average pairwise Euclidean distance within each group in the latent space ($\pm 2\sigma$ in blue). An F-test is performed to compute a p-value on the decrease of the variance [a] for each preterm category compared to full-term. $(, 28)$: $\mathbf{p = 0.01}$, $(28, 32)$: $\mathbf{p = 0.01}$, $(32, 37)$: $p = 0.1$. **c)** STS visualization (red) in whole brain context (sulci in blue) of dHCP subjects sampled based on the extremely preterm classification score. In each group, 19 subjects are sampled for the long knob pattern identification (marked in green). Only 4 are shown here, Laval (2024) gives exhaustive visualization.

[a] computed as the average squared Euclidean distance to the barycenter of the distribution.

## 4. Conclusion

In this work, using SimCLR, we identify an STS pattern absent from the extremely preterm data, suggesting a more unique developmental trajectory for preterms than full-terms. We showcase the transferability of the model optimized on adults in the cingular region to baby data in the STS region. Future work will be to repeat such analysis in all brain regions for the three classes of prematurity to uncover other regions altered by preterm birth.

## Acknowledgments

This work was funded by the ANR grants FOLDDICO and BABYTOUCH.

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
