# OpenReview forum: "Self-supervised contrastive learning unveils cortical folding pattern linked to prematurity"
_MIDL.io/2024/Short_Papers — MIDL 2024 Short Papers_

### Official Review · Reviewer_EzcV · 2024-04-24

**Confidence:** 5
**Final Rating:** 4

**Review:**

*** 111 Self-supervised contrastive learning unveils cortical folding pattern linked to prematurity
The submissions proposes to use self-supervised contrastive learning (SimCLR) to analyze brain folding patterns in the superior temporal sulcus region of preterm babies. The results find that the model could distinguish between extremely preterm, very preterm, moderately preterm, and full-term babies. These results suggest that this method has the potential to reveal unique developmental trajectories in preterm brains. However, more work is needed to further validate the findings. The methodological details may currrently be insufficient to assess the feasibility of the approach. Additionally, the results are very coarse, based on only 3 time points.  Despite these limitations, the potential conclusions are worthy of publication as the work could foster constructive discussions in the field. Authors are encouraged to further the investigations. The findings could be presented in a poster format to stimulate discussion and collaboration among researchers. For these reasons, the recommendation is towards Acceptance.

---

### Decision · Program_Chairs · 2024-04-26

Accept